# Neutrophils in *Mycobacterium tuberculosis*

**DOI:** 10.3390/vaccines11030631

**Published:** 2023-03-12

**Authors:** Cheldon Ann Alcantara, Ira Glassman, Kevin H. Nguyen, Arpitha Parthasarathy, Vishwanath Venketaraman

**Affiliations:** 1Department of Basic Sciences, College of Osteopathic Medicine of the Pacific, Western University of Health Sciences, Pomona, CA 91766, USA; 2Division of Biology, Pitzer College, Claremont, CA 91711, USA

**Keywords:** neutrophils, tuberculosis, infection, *Mycobacterium tuberculosis*, immune system

## Abstract

*Mycobacterium tuberculosis* (*M. tb*) continues to be a leading cause of mortality within developing countries. The BCG vaccine to promote immunity against *M. tb* is widely used in developing countries and only in specific circumstances within the United States. However, current the literature reports equivocal data on the efficacy of the BCG vaccine. Critical within their role in the innate immune response, neutrophils serve as one of the first responders to infectious pathogens such as *M. tb*. Neutrophils promote effective clearance of *M. tb* through processes such as phagocytosis and the secretion of destructive granules. During the adaptative immune response, neutrophils modulate communication with lymphocytes to promote a strong pro-inflammatory response and to mediate the containment *M. tb* through the production of granulomas. In this review, we aim to highlight and summarize the role of neutrophils during an *M. tb* infection. Furthermore, the authors emphasize the need for more studies to be conducted on effective vaccination against *M. tb*.

## 1. Introduction

Globally, *Mycobacterium tuberculosis* (*M. tb*) infects over 10 million people per year. As the 13th leading cause of death and second leading infectious killer in 2021, *M. tb* bears a severe burden despite the discovery of the bacilli in 1882 and the use of the Bacillus Calmette-Guérin (BCG) vaccine starting in 1921 [1,2]. *M. tb* results in asymptomatic clearance of the pathogen, clinical infection of varying severity, or latent tuberculosis infection (LTBI) with the latter being more common. Clinical tuberculosis infection may cause symptoms of malaise, cough, chest pain, hemoptysis, and fever, whereas LTBI is typically asymptomatic [3]. LTBI arises as the body protects itself by restricting the pathogen’s replication, spread, and oxygen supply [4]. The greatest dangers of LBTI are in reactivation to active TB and subsequent silent spread [5]. An immunocompromised state can induce liquefaction of the caseum in the granuloma and bacterial replication, thereby promoting cavity formation and intrapulmonary spread of *M. tb* [6]. There is an increased risk of reactivating *M. tb* in individuals whose immune systems are weak, such as those with human immunodeficiency virus (HIV) and diabetes mellitus (DM) [7,8]. Of note, the amount of neutrophils present is decreased in these chronic infections [9,10]. Neutrophils make up the largest amount of white blood cells in the human body and are significant parts of the body’s defense system against *M. tb*. In the process of acute inflammation, neutrophils perform several specialized functions, including chemotaxis, phagocytosis, generation of reactive oxygen species (ROS), and production of neutrophil extracellular traps [11]. However, the role of neutrophils in active TB infection compared to LBTI is highly controversial and contradictory. The cytokine IL-17 is particularly important for neutrophil recruitment to the lungs. Mice with IL-17R signaling knocked out showed a significant delay in neutrophil recruitment into the alveolar space, followed by decreased survival against bacterial challenge [12]. Recently, a new vaccine called *M. smegmatis*-Ag85C-MPT51-HspX (mc2-CMX) has been shown to protect mice against *M. tb* challenge, via a neutrophil-mediated response of specific Th1 and Th17 cells [13]. Another vaccine containing nanoemulsion (NE)-based adjuvants delivered with *M. tb* specific immunodominant antigens (NE-TB vaccine) has been shown to induce potent mucosal IL-17 T-cell responses. Here we highlight the nuanced, critical role of neutrophils in *M. tb* infection to highlight the need for further research of these immune cells in additional vaccines.

## 2. Pathogenesis of TB

*M. tb* infection occurs via the inhalation of aerosolized droplet nuclei containing viable bacilli that travel down the respiratory tract before eventually localizing to the lungs’ alveoli [14]. Mycobacteria-laden droplet nuclei can remain suspended in the air for several hours after a patient with active pulmonary TB coughs, sneezes, speaks, or sings [4,15]. After settling in the lung alveoli, *M. tb* can replicate intracellularly and travel throughout the body by way of the blood and lymph to infect other organ systems. Within 2–8 weeks of initial infection, specialized immune cells work to phagocytose the pathogenic bacilli. If host defenses are not able to fully eliminate the bacilli, but contain the bacilli so minimal clinical symptoms manifest, then a patient is classified to have LTBI [5,16]. If immune cells fail to eliminate or contain the bacilli, this allows for growth and induction of clinical symptoms. Fewer than 10% of infected individuals develop primary, or active, TB with clinical symptoms [5].

Granulomas, the pathologic structure most closely associated with TB, serve as a reservoir of infection [16]. *M. tb* can reside in these structures for decades and progress to active TB at any point [17]. Aside from using granulomas as a protective shelter, the mycobacteria have a variety of other methods to escape the defenses of the immune system. These include inhibition of the maturation and acidification of phagolysosomes and inhibition of apoptosis and autophagy [18,19]. Once the pathogen is internalized in a phagosome, the ESX-1 secretion system plays a major role in the pathogenesis of active TB infection by facilitating the delivery of bacterial products into the cytoplasm of macrophages [20].

## 3. Role of Neutrophils in the Immune Response

Neutrophils are produced in the bone marrow from hematopoietic stem cells which follow myeloid lineage differentiation. Their differentiation is controlled by granulocyte colony stimulating factor (G-CSF). Neutrophils are held in the bone marrow by the binding of CXCL12, a ligand expressed by osteoblast cells, to CXCR4, a receptor on neutrophils. G-CSF downregulates CXCL12 and CXCR4. As neutrophils approach maturation, CXCR4 is downregulated while CXCR2 and TLR4 are upregulated. Conserved TLRs play a key role in recognition and host resistance to *M. tb*, with TLR4 and TLR2 expression in neutrophils significantly increased in TB patients [21]. To exit the bone marrow, endothelial cells in vasculature express the CXCL8 ligand to bind to CXCR2 for mobilization into the bloodstream [22]. CXCL8 is also secreted by activated macrophages during *M. tb* infection which is critical for neutrophil recruitment [23]. Neutrophil CXCR2 expression is also increased in TB patients and together, these factors serve to upregulate the mobilization of neutrophils in *M. tb* infection [24]. 

Neutrophils are the most abundant granulocyte, comprising 50–70% of all circulating leukocytes, and the first responders of the innate immune system in infection. Increased circulating interleukin-1 (IL-1), released by macrophages during *M. tb* infection and other inflammatory states, can stimulate neutrophil production via the IL-17-G-CSF axis [25]. Neutrophils possess two primary mechanisms of pathogen clearance: phagocytosis for intracellular destruction of the pathogen and degranulation for extracellular destruction [26]. Neutrophils utilize NADPH oxidase to produce reactive oxygen species (ROS) inside their phagosome for pathogen destruction, as well as extracellular ROS in response to soluble agonists [27]. Degranulation follows a tightly controlled release of 4 major types of granules: primary or azurophilic, secondary or specific, tertiary or gelatinase, and secretory. Primary granules contain myeloperoxidase (MPO), cathepsin G, elastase, proteinase 3, and defensin, responsible for pathogen eradication. Defensins exhibit anti-mycobacterial activity and have been investigated as an alternate therapeutic option for tuberculosis treatment [28]. Secondary granules contain lactoferrin, responsible for sequestering iron, copper, and proteins to reduce pathogen growth. Tertiary granules contain gelatinase proteins, such as matrix metalloproteinase-8 (MMP-8), capable of a wide range of functions including activation of IL-1B. Lastly, secretory granules contain albumin and cytokines [29,30]. Tan et al. demonstrated how phagocytosis of apoptotic neutrophils by macrophages during active *M. tb* infection led to destruction of intracellular *M. tb* by trafficking of granule contents. This finding represents a bridge of defense between neutrophils and macrophages in response to *M. tb* intracellular infection [31]. 

Neutrophils are also able to participate in the adaptive immune response. They can aid in B cell activation and survival by the production of cytokines B cell activating factor (BAFF) and a proliferation-inducing ligand (APRIL). Neutrophils can also serve as anti- and pro-inflammatory modulators for T cells. Arginase-1, present in primary granules, as well as ROS release, can decrease T cell activation and proliferation. They can also serve as antigen-presenting cells by upregulating their MHC-II levels during IFN-gamma stimulation, along with costimulatory molecules, to induce Th1 and Th17 differentiation [29,30]. Th1 cells secrete IFN-gamma which upregulate macrophage response to control *M. tb* infection, whereas Th17 cells induce neutrophilic inflammation and tissue damage, serving as a mediator of *M. tb* pathology [32]. Neutrophils play a crucial role in the activation and expression of adaptive immune responses.

Localization of neutrophils to tissues is important for their function. One mechanism by which neutrophils are localized is through aging. As neutrophils age, they express a greater abundance of CXCR4. In humans, there exists a large pool of aged neutrophils in the vascular lumen and interstitial space of the lungs, thought to be retained by CXCR4-dependent mechanism [22]. Likewise, other subpopulations of neutrophils find their way to specific regions of the body, such as CCR7 and integrin LFA-1 expressing neutrophils which are directed to lymph nodes [22]. Interestingly, one subpopulation of neutrophils, phenotype CD49dhi CXCR4hi VEGFR1, are attracted to hypoxic tissue and may represent one mechanism by which neutrophils are drawn to the hypoxic TB granuloma [22,25,33]. While neutrophils were originally dismissed as having a role in chronic TB due to their short lifespan, evidence suggests they play a key role in granuloma formation in chronic TB [33,34]. There is evidence that neutrophil lifespan is increased in hypoxic conditions, and even potentiate hypoxic conditions as a positive feedback loop through their consumption of oxygen [25]. A summary of the role of neutrophils is represented below in Figure 1. More evidence of the role of neutrophils in acute and chronic TB infection will be presented in more detail in the next sections. 

There is evidence that neutrophils can specifically be primed by microbiota and pathogens [22,26]. Neutrophils are strongly activated by *M. tb* infection. They are the primary immune cell type in cavities of pulmonary TB patients, associated with MMP8-mediated extracellular matrix destruction and cavity formation. In addition to contributing to the pathogenesis of TB, MMP8 is also associated with the release of neutrophil extracellular traps (NETs). NETs are released by a process called NETosis as a combination of DNA, histones, and antimicrobial granule proteases released to entrap *M. tb* and will be discussed in more detail later [35,36].

## 4. Neutrophils and Extracellular Traps

Upon stimulation by physiological states, damage-associated molecular patterns, or microbial pathogen-associated molecular patterns, neutrophils undergo NETosis to release NETs. NETs are a conglomerate of decondensed DNA strands, histones, and antimicrobial peptides and proteins, including granule constituents such as myeloperoxidase and elastase. NETs function to promote pathogen clearance by promoting phagocytosis by macrophages, entrap extracellular pathogens to prevent infection spread, and have been implicated in acute and chronic inflammatory disorders and microbial pathogenesis, namely *M. tb* pathogenesis [26,34,36].

NETs formation begins with the production of oxidants by neutrophils which leads to degranulation of the nuclear envelope and release of DNA into the cell. Two enzymes are responsible for regulating the degranulation and decondensing of DNA, peptidyl arginine deiminase type IV (PAD4), and neutrophil elastase. PAD4 facilitates conversion of positively charged arginine side chains into neutral side chains on histones through the process of citrullination. Citrullination decondenses chromatin to facilitate DNA release. DNA then serves as a negatively charged scaffold for NET components, such as histones and proteases, to bind via electrostatic charge. While initially thought to require lysis for NETosis, studies have found that NETs can be released through vesicular transport [29]. 

*M. tb* is capable of stimulating NET synthesis [35]. Francis et al. demonstrated that the SAT-6 protein, secreted by *M. tb* in early infection, induces production of NETs [37]. In addition, Dang et al. identified an extracellular factor of *M. tb*, spingomyelinase Rvo888, which is capable of inducing NETs formation and enhancing colonization ability of recombinant *M. smegmatis* in the lungs of mice [38]. Van de meer et al. revealed that patients with active TB had elevated plasma levels of nucleosomes and elastase, biomarkers for NET formation, when compared with healthy participants [39]. Elevated serum levels of citrullinated histone H3, another NET biomarker, is associated with lung cavitation and poor treatment outcomes [40]. As such, NET production may serve as a method to monitor TB severity, as severity of TB infection correlates positively with TB patient’s plasma NET levels [41]. Braian et al. found that *M. tb*-induced NETs contained Hsp72, a protein which can trigger cytokine release from macrophages, and may play a role in the interaction between neutrophils and macrophages during acute phase of *M. tb* infection [42]. Moreira-Teixeira et al. revealed that type 1 interferon (IFN-1) signaling induces pulmonary NETosis and promotes mycobacterial growth [40]. They also showed that IFN-1-induced NETosis is associated with infection severity in TB-susceptible mice. 

Evidence shows that while *M. tb* induced NETs formation, and successfully captured *M. tb*, the NETs and neutrophils were unable to kill *M. tb*. It is suggested that NETs can capture *M. tb* due to *M. tb*’s electron-dense, polysaccharide-containing negatively charged outer layer electrostatic attraction to the positively charged NET structure [34]. It has been suggested that NETs provide a foundation for *M. tb* replication which promotes cavitation growth. As the granuloma in TB is a hypoxic environment, it is important to understand the role of NETosis after granuloma formation has been completed. Ong et al. found that in hypoxic conditions, NET formation, neutrophil apoptosis, and neutrophil necrosis were all inhibited, whereas the secretion of NET components such as MMP9 and elastase were increased. While NETs play a role in cavitation and infection severity, they appear to serve less of a role within the granuloma stage of infection [33].

Recently, Su, et al. revealed that a subpopulation of neutrophils called tuberculosis-related low-density granulocytes (LDGs) release abnormally high levels of NETs. The high level of NETs produced during *M. tb* infection triggered a shift from normal-density granulocytes (NDGs) to more LDGs, whereas inhibition of NET release prevented their conversion. In addition, they showed that reducing ROS production significantly decreased the conversion of NDGs to LDGs as well [43]. This provides evidence that *M. tb* may induce a shift from NDGs to LDGs via ROS pathway, thus increasing NETs production. Deng et al. further revealed that LDGs are associated with the severity of TB infection [44]. Rao et al. found that LDGs can inhibit the production of interferon-gamma in T cells, both decreasing the efficacy of the T-SPOT TB assay used for detection of *M. tb* and modulating the macrophage response [45]. The exact role LDGs play in TB infection requires further investigation.

## 5. Neutrophils in the Acute Stage of *M. tb*

Neutrophils are the main cell type during active TB infections, and the first phagocytes recruited from the pulmonary vasculature to the pulmonary interstitium via chemotaxis initiated by alveolar macrophages [46,47]. As mentioned previously, the cytokine IL-17 is known to recruit neutrophils in the airway [48,49]. Mechanisms such as phagocytosis, degranulation, ROS formation, and NET release are employed by the neutrophils to combat *M. tb*. Macrophages, dendritic cells, natural killer cells, fibroblasts, CD4 T cells, and cytotoxic CD8 T cells are also recruited to the infection site via cytokine secretion, leading to further containment of the bacterium [47]. 

Neutrophils are a critical component in response to early TB infection, as established by multiple animal studies. For example, Sugawara et al. found that lipopolysaccharide (LPS) induced neutrophilia prevented early *M. tb* infection in rats. If neutrophilia was induced 10 days after aerial infection with *M. tb*, neutrophils were not able to prevent development of infection [50]. This highlights that the presence of neutrophils seems to have a significant impact in combatting infection. Another study by Pedrosa et al. found that the depletion of neutrophils during the first week infection exacerbates *M. tb* proliferation in mice, specifically in the liver, spleen, and lung. Rather than directly eliminating *M. tb*, it was suggested that the protective nature of the neutrophil was mediated via a nonphagocytic, possibly immunomodulatory, mechanism that affected IFN-gamma production [51]. Yang et al. conducted a study on neutrophils in mycobacterial infection using a zebrafish transgenic line with fluorescently labeled neutrophils. Initially, neutrophils were found to be recruited to granulomas by signals from dying, infected macrophages. Then the neutrophils phagocytosed the infected macrophages and killed their phagocytosed mycobacteria via oxidative mechanisms [52]. This study not only shows the role of neutrophils in acute mycobacterial infection, but also corroborates the suggestion of Petrosa et al. that these immune cells indirectly participate in the elimination of mycobacteria. 

## 6. Neutrophils in Latent Phases of *M. tb* Infection

The complete role of neutrophils in the development and maintenance of *M. tb* infection has not been well characterized, in comparison to other types of immune cells such as macrophages and CD4+ T cells. Neutrophils play a complex role throughout the different phases of *M. tb* infection, ranging from protective to pathological. Granuloma formation, with its oxygen-depleted environment, is a defining characteristic of TB [4]. Specifically, Barry et al. defines a granuloma as an organized structure comprised of neutrophils, lymphocytes, macrophages, and sometimes fibroblasts, often with a necrotic center [53]. Granulomas may be present in both active and latent TB infections. The formation of a granuloma alone is insufficient to eradicate infection. Instead, proper functioning of a granuloma determines the ultimate outcome of infection [54]. Regarding LTBI, granulomas serve as a reservoir for reactivation. Hypoxia initiates the angiogenesis required for the formation of granulomas [33]. Here we examine recent studies regarding the role of neutrophils within these hypoxic granulomas. 

As discussed earlier, neutrophils are essential parts of the immune system. It is recognized that neutrophils are present in granulomas and aid in their formation [46]. Neutrophilic phagocytosis is important against *M. tb* to produce a pro-inflammatory response and recruit more immune cells. As a result, the timely removal of spent neutrophils is just as important. They need to be regulated through apoptosis and clearing by macrophages to avoid necrotic lysis to the surrounding tissue [55]. Neutrophils were found to contribute to the cytokine milieu in granulomas, and express cytokines in more granuloma microenvironments than T cells [56]. Thus, this highlights the important role of neutrophils as immunoregulatory cells in LTBI. The general views are that neutrophils play a protective role in early granulomas by working to eliminate *M. tb*, but a pathological role in late granulomas by damaging surrounding tissue. 

The response of neutrophils to hypoxia within granulomas has been studied more in the past few years. Ong et al. showed that the hypoxic environment of the granulomas exacerbates neutrophil-dependent immunopathology in TB infection. In response to hypoxic conditions, neutrophils were found to increase secretion of enzymes such as neutrophil MMP-8, MMP-9, and elastase to drive matrix destruction [33]. This further supports a previous study that found hypoxia increases neutrophil collagenase expression to destroy lung tissue [57]. Hypoxia has been found to increase neutrophil life span, reactive nitrogen species, protease secretion, and destruction of collagen and elastin. On the other hand, hypoxia decreases apoptosis [58,59,60]. This is detrimental in prolonging the inflammatory response that neutrophils produce, which may further exacerbate tissue damage [61]. 

Animal and whole blood studies suggest a positive correlation between severity of TB and neutrophil abundance. For example, neutrophil depletion in mouse models reduced lung pathology and bacterial burden during TB infection [62,63]. *M. tb* entry sites, infection progress, and infectious load may result in differed effects of neutrophils. The abundance of neutrophils increases in the lungs as TB progresses and results in more severe pathology in guinea pig, mouse, and macaque models [56,64,65]. This finding is also supported in whole blood samples [66]. In contrast, studies have also shown a protective role for neutrophils during *M. tb* infection, particularly in blood. Neutrophils restricted the growth of *M. tb* in human whole blood ex vivo [67]. Overall, neutrophils play a role in the progression of TB infection, with prognosis dependent on neutrophil viability, infection severity, and location of infection (pulmonary or systemic). The “Trojan horse” phenomenon, where mycobacteria utilize neutrophils to their own benefit—whether for nutrients or shelter—has been considered [62,64,68,69]. This may be the case more commonly in more severe infections, especially when neutrophils have not been properly cleared by macrophages. In sum, neutrophils are a significant part of the immune response to *M. tb*. In early infection, rapid phagocytosis, degranulation, and ROS work to combat the pathogen. In later infection, due to hypoxia and/or insufficient clearing, correlates to poor infection outcomes and more severe pathology. Given that neutrophils are the predominant immune cell type present in active TB patients and are present in LTBI, understanding their roles and how they change over the course of infection is essential to develop current treatments against *M. tb*. The role of neutrophils during active and LTBI is summarized in Figure 2 below.

## 7. Using Neutrophils for Good: Vaccine Development

Effective eradication of *M. tb* has been unsuccessful. Currently, the bacilli Calmette-Guerin vaccine (BCG) is the only licensed TB vaccine. The BCG is a live, attenuated vaccine that is commonly used to prevent TB meningitis in children in developing countries. In the United States, the vaccine is only administered under specific circumstances, such as children who have a negative tuberculin skin test (TST) and are continually exposed to adults who are untreated or ineffectively treated for TB infection or have isoniazid and rifampin resistant *M. tb*. In addition, healthcare workers in settings with a high percentage of TB patients infected with isoniazid and rifampin resistant *M. tb* strains are considered to receive BCG vaccination [70]. However, this vaccine is generally not recommended in the United States due to its limited efficacy in adults and ability to produce a false positive reaction to the TST [68]. 

The natural route of TB infection is through the mucosal surface, yet no mucosal TB vaccine is currently in clinical trials. Ahmed et al. discuss the use of a more effective vaccine that is delivered through the mucosal route. The NE-TB vaccine induced potent mucosal IL-17 cell responses to protect against *M. tb* infection, and when delivered along with BCG, decreased severity of infection [71]. The requirement of IL-17R signaling for neutrophil recruitment, host defense, and G-CSF expression highlights the importance of this pathway against *M. tb* infection [12].

Another vaccine called *M. smegmatis*-Ag85C-MPT51-HspX (mc2-CMX) has been shown to protect mice against *M. tb* challenge, through a neutrophil-mediated response of specific Th1 and Th17 cells. In comparison to BCG, the mc2-CMX induced a superior humoral and cellular response in mice exposed to *M. tb* [13]. Trentini et al. studied the effect of neutrophil depletion of immune response in mice that were vaccinated with mc2-CMX. Mice whose neutrophils were depleted during vaccination, when challenged with *M. tb*, did not show an increase in the Th1 and Th17 response seen among neutrophils sufficient groups. This suggests that the absence of neutrophils during mc2-CMX vaccination C57BL/6 mice reduces protection and results in the absence of a Th1 and Th17 recall response. This study emphasizes that neutrophils are important for the induction of a CMX Th1-specific response against *M. tb*, due to their relationship with Th1 and Th17 cells [72]. As neutrophils are the most common type of white blood cell and have been play a part in invoking an enhanced immune response in mice vaccinated with mc2-CMX when compared to the current vaccine strategy, this emphasizes the need for further research directed towards the function of neutrophils in additional vaccines.

## 8. Conclusions

*M. tb* continues to burden the health and economy of developing nations. The need for effective preventative practices is pivotal to lower the incidence and prevalence of cases throughout the world. More research is needed on emerging vaccines to establish the efficacy of promoting a neutrophil-mediated response of specific Th1 and Th17 cells within the host. As the most abundant white blood cell, neutrophils play a vital role in the immune response to *M. tb*. During an active infection, neutrophils migrate from the bloodstream via chemotaxis to deploy various means of liberating *M. tb* from the host. These include phagocytosis, production of ROS, and degranulation of destructive enzymes to accentuate *M. tb* clearance. Furthermore, neutrophils recruit macrophages, NK cells, B and T cells through the secretion of cytokines. During LTBI, neutrophils crucially contain *M. tb* through the production of granulomas to prevent further spread. As new research on *M. tb*. evolves, augmentation of the robust immune response by neutrophils may serve as a promising avenue to investigate. 

## Figures and Tables

**Figure 1 vaccines-11-00631-f001:**
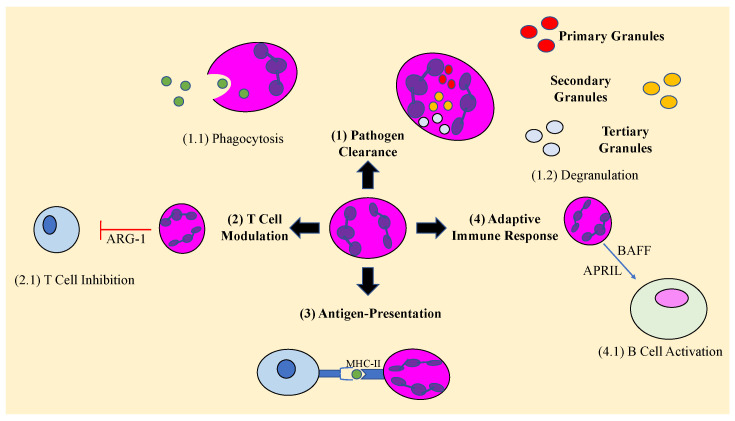
Neutrophil-mediated roles during an immune response. Arginase 1 (ARG-1), major histocompatibility complex II (MHC-II), B-cell activating factor (BAFF), A Proliferation-Inducing TNF Ligabd (APRIL).

**Figure 2 vaccines-11-00631-f002:**
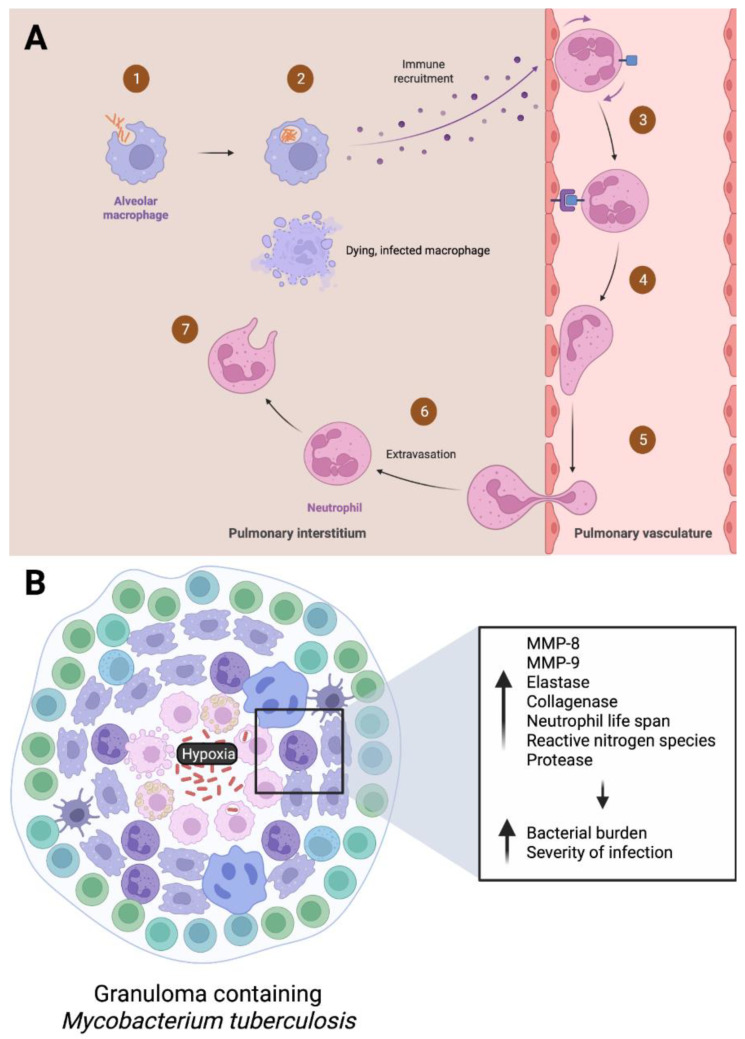
Neutrophil response during an active and latent *M. tb*. infection. During an (**A**) active infection, alveolar macrophages encounter, and phagocytose *M. tb*. This results in chemokine release and initiation of margination, rolling, adhesion, and extravasation of neutrophils to the infected site. If the macrophages are unable to eliminate the pathogen themselves, neutrophils may phagocytose the dying, infected macrophages. (**B**) LTBI commonly occurs via granuloma formation, where the immune system works to contain *M. tb*. Hypoxic conditions within granulomas have been shown to increase the secretion of enzymes matrix metalloproteinase 8 and 9 (MMP-8, MMP-9), elastase, collagenase, and proteases as well as increase neutrophil life span and reactive nitrogen species. Unregulated neutrophil inflammation contributes to increased bacterial burden and severity of TB infection.

## Data Availability

Not applicable.

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
