# Peer review of "Neutrophils in Mycobacterium tuberculosis"

_vaccines, 2023, doi:10.3390/vaccines11030631_

Round 1

Reviewer 1 Report

In the review article, Alcantara et al. discussed the role of neutrophils during Mtb infections. Overall, the sections are comprehensive and logically structured. However, the review article contained numerous inaccuracies in the research findings. The use of professional English and the quality of writing also need improvement. 

Major comments:

  1. There are a few mistakes in the manuscript. For example, in lines 62-63, the authors attribute LBTI to the successful enclosure of Mtb within the granuloma, which lacks literature support. 
  2. The use of professional English needs improvement. The article contains a few grammar mistakes, such as lines 27-28, and 205-207, that need correction. The authors can also improve the readability of the paragraphs by including more connective sentences. For example, there is a jump from neutrophils to IL17 in line 43. Furthermore, in lines 162-186, the authors piled evidence without building connections or addressing the 'so what?' Without tight connections between sentences, I often have trouble identifying each paragraph's focus. 
  3. In section 3 on neutrophils, the authors wrote three paragraphs on the general functions of neutrophils without reference or connection to tuberculosis. I suggest focusing the discussion on tuberculosis.

Minor comments: 

  1. In lines 200-201, the authors mentioned 'fast and slow-replicating Mtb.' I am not sure what it means and need further clarification.
  2. In line 234, the authors mentioned a positive correlation between neutrophils and TB severity. Supporting evidence is needed.
  3. In lines 280-283, it is hard to understand how similar bacterial load in neutrophil depleted and non-depleted and non-vaccinated leads to the conclusion of the protective nature of neutrophils. Please clarify. 

Author Response

Reviewer #1

Comments and Suggestions for Authors

In the review article, Alcantara et al. discussed the role of neutrophils during Mtb infections. Overall, the sections are comprehensive and logically structured. However, the review article contained numerous inaccuracies in the research findings. The use of professional English and the quality of writing also need improvement.

Response: We would like to thank you for your thorough review. We appreciate all the attention to detail and suggestions offered in order to increase the quality of our paper.

Major comments:

  1. There are a few mistakes in the manuscript. For example, in lines 62-63, the authors attribute LBTI to the successful enclosure of Mtb within the granuloma, which lacks literature support.

Response: Thank you for your feedback. We agree that this assertion has not been definitively found, therefore we have altered the writing to show that granulomas are present commonly in LTBI in an attempt to encapsulate and contain the pathogen.

  1. The use of professional English needs improvement. The article contains a few grammar mistakes, such as lines 27-28, and 205-207, that need correction. The authors can also improve the readability of the paragraphs by including more connective sentences. For example, there is a jump from neutrophils to IL17 in line 43. Furthermore, in lines 162-186, the authors piled evidence without building connections or addressing the 'so what?' Without tight connections between sentences, I often have trouble identifying each paragraph's focus.

Response: Thank you for your feedback. We have since corrected the grammatical errors and added more explanations for why studies were included and their connections with each other.

  1. In section 3 on neutrophils, the authors wrote three paragraphs on the general functions of neutrophils without reference or connection to tuberculosis. I suggest focusing the discussion on tuberculosis.

Response: Thank you for your feedback. We have narrowed down the section to focus on tuberculosis, specifically with connecting sentences for why certain functions are significant during infection.

Response: Thank you for your feedback. We agree further clarification was needed. There was a discrepancy between what we wrote and what the paper was illustrating, which we have corrected. We have provided details on the study and how this pertains to our paper.

  1. In line 234, the authors mentioned a positive correlation between neutrophils and TB severity. Supporting evidence is needed.

Response: Thank you for your feedback. We mentioned this at the beginning of the paragraph as an opening sentence to be discussed with all the evidence mentioned after. We realize this may not have been clear, so we have adjusted the writing accordingly.

  1. In lines 280-283, it is hard to understand how similar bacterial load in neutrophil depleted and non-depleted and non-vaccinated leads to the conclusion of the protective nature of neutrophils. Please clarify.

Response: Thank you for your feedback. We agree further clarification was needed. We have since included more details on the study and how this pertains to our paper.

Reviewer 2 Report

Neutrophils in Mycobacterium tuberculosis infection

Dear author and editor,

The review covered the topic and well written. The article could be published after a minor revision.

 I have just one comment : the relation or the cooperation between neutrophils and macrophages could be further explained

Thank you, best regards

Author Response

Reviewer #2

Comments and Suggestions for Authors

The review covered the topic and well written. The article could be published after a minor revision.

I have just one comment: the relation or the cooperation between neutrophils and macrophages could be further explained.

Response: Thank you for your feedback. We have included more information within the text as well as within Figure #2 to show how alveolar macrophages first encounter M. tb, then work to recruit neutrophils to the site of infection and how neutrophils have been found to phagocytose the dying, infected-macrophages if sufficient elimination does not occur.

Reviewer 3 Report

A well-written and comprehensive review on the role of neutrophils in M. tuberculosis infection. A main focus is on the role of NETs during infection, whereas the role of apoptotic cells and efferocytosis is more or less lacking. Since neutrophils are short lived and apoptotic cells and bodies are  present in the granulomas, data related to apoptotic neutrophils and their role during TB pathogenesis should be discussed.

Figure 2 is too schematic and not very informative in its present form. Should include more detailed information.

Author Response

Reviewer #3

Comments and Suggestions for Authors:

A well-written and comprehensive review on the role of neutrophils in M. tuberculosis infection. A main focus is on the role of NETs during infection, whereas the role of apoptotic cells and efferocytosis is more or less lacking. Since neutrophils are short lived and apoptotic cells and bodies are present in the granulomas, data related to apoptotic neutrophils and their role during TB pathogenesis should be discussed.

Response: Thank you for your feedback. We have expanded upon neutrophils in granulomas, especially how hypoxia increases neutrophil life span (instead of undergoing apoptosis) to contribute to more severe TB infection.

Figure 2 is too schematic and not very informative in its present form. Should include more detailed information.

Response: Thank you for your feedback. We have redone the entire figure to further illustrate the important points from acute TB infection and LTBI. We include the relationship of macrophages and neutrophils as well as visualize the effect of hypoxic conditions on neutrophils within granulomas.

Round 2

Reviewer 1 Report

The authors have addressed most of my concerns and made significant improvements to the manuscript.